# Enhanced Electrocatalytic Water Oxidation of Ultrathin Porous Co_3_O_4_ Nanosheets by Physically Mixing with Au Nanoparticles

**DOI:** 10.3390/nano12244419

**Published:** 2022-12-11

**Authors:** Changhe Hu, Dejuan Sun, Jie Liu, Qi Zhang, Xiao Li, Huhui Fu, M. Liu, Jiayue Xu, Guojian Jiang, Yalin Lu

**Affiliations:** 1School of Materials Science & Engineering, Shanghai Institute of Technology, Shanghai 201418, China; 2Department of Materials Science and Engineering, University of Science and Technology of China, Hefei 230026, China; 3Hefei National Laboratory for Physical Sciences at the Microscale, Hefei 230026, China

**Keywords:** Co_3_O_4_/Au, composite, OER, XANES, electronic structure

## Abstract

Ultrathin porous Co_3_O_4_ nanosheets are synthesized successfully, the thickness of which is about three unit-cell dimensions. The enhanced oxygen evolution reaction (OER) performance and electronic interaction between Co_3_O_4_ and Au is firstly reported in Co_3_O_4_ ultrathin porous nanosheets by physically mixing with Au nanoparticles. With the loading of the Au nanoparticles, the current density of ultrathin porous Co_3_O_4_ nanosheets is enhanced from 9.97 to 14.76 mA cm^−2^ at an overpotential of 0.5 V, and the overpotential required for 10 mA cm^−2^ decreases from 0.51 to 0.46 V, smaller than that of commercial IrO_2_ (0.54 V). Furthermore, a smaller Tafel slope and excellent durability are also obtained. Raman spectra, XPS measurement, and X-ray absorption near edge structure spectra (XANES) show that the enhanced OER ascribed to a higher Co^2+^/Co^3+^ ratio and quicker charge transfer due to the electronic interaction between Au and ultrathin Co_3_O_4_ nanosheets with low-coordinated surface, and Co^2+^ ions are beneficial for the formation of CoOOH active sites.

## 1. Introduction

Water electrolysis is a promising approach to store and supply clean and sustainable sources of energy [1,2], the development of which is driven by the dwindling energy supplies and environmental pollution [3,4]. Unfortunately, its efficiency and commercial implementation are severely restricted due to sluggish oxygen evolution reaction (OER) kinetics, which are of the thermodynamic uphill reaction that involves complex four-electron process [5,6], and the formation of O-O bond needs to overcome the high energy barrier, which means that a high overpotential is required to drive the reaction. Owing to the scarcity and high cost of current catalysts, e.g., IrO_2_ and RuO_2_ [7,8], Co_3_O_4_ is developed as a replacement for noble-metal-based catalysts, which is low-cost, abundant, and has high chemical stability. However, it still suffers from low electron conductivity, poor cycling stability, and relatively high overpotential [9,10,11,12].

The current research demonstrates that ultrathin nanosheets with limited atomic layer stacking possess better electrocatalytic water oxidation than bulk materials due to their large surface-to-volume ratio and abundant active sites. Xie’s group designs NiCo_2_O_4_ nanosheets with ultrathin thickness, which show a higher current density and lower overpotential and larger turnover frequency (TOF) than the bulk sample, and the authors attribute them to the synergistic effects between the oxygen vacancies and ultrathin thickness, which could lead to a high interfacial contact area with the electrolyte, short ion diffusion paths, and fast adsorption of the H_2_O molecules. Subsequently, the OER process is accelerated [13]. Recently, as a highly efficient catalyst for the OER, graphene-like holey Co_3_O_4_ nanosheets were reported, which exhibit low onset potential of 0.617 V vs. Hg/HgO, a high current density of 12.26 mA cm^−2^ at 0.8 V vs. Hg/HgO, and long-term stability with negligible fading in current density after 2000 cycles [14]. On the other hand, catalyst supports are also important to OER activity. Yeo and Bell’s research shows that Au-supported Co_3_O_4_ film exhibits better OER performance than other noble metal-supported catalysts [15]. However, such enhancement decreased rapidly with the increase in the catalyst’s thickness, leading to low catalyst loadings and limited enhancement [15,16,17]. Herein, developing ultrathin nanosheets catalysts with the beneficial effect of Au could be an effective strategy to increase catalyst loading and improve OER performance. To the best of our knowledge, there are few reports on the Au-enhanced OER performance of Co_3_O_4_ ultrathin nanosheets; furthermore, such a promoting effect does not exist in the physically mixed Au/catalysts.

Here, ultrathin porous Co_3_O_4_ nanosheets are synthesized successfully by a facile method, and the corresponding OER performance is firstly enhanced by physically mixing Co_3_O_4_ ultrathin porous nanosheets with Au nanoparticles. With the loading of Au nanoparticles, the overpotential at a current density of 10 mA cm^−2^ decreases from 0.51 to 0.46 V, both of which are smaller than that of commercial IrO_2_ (0.54 V), and the current density of ultrathin porous Co_3_O_4_ nanosheets is enhanced from 9.97 to 14.76 mA cm^−2^ at an overpotential of 0.5 V. Furthermore, Co_3_O_4_/Au nanocomposites possess a smaller Tafel slope (88 mV dec^−1^) than Co_3_O_4_ ultrathin nanosheets (100 mV dec^−1^). X-ray photoelectron spectroscopy (XPS), Raman spectra, and XANES spectra indicate that the enhanced OER performance is ascribed to the higher Co^2+^/Co^3+^ ratio and quicker charge transfer in Co_3_O_4_/Au nanocomposites than that in the ultrathin porous Co_3_O_4_ nanosheets, and Co^2+^ ions benefit from the formation of CoOOH, which serves as an active site for the OER.

## 2. Experimental Section

### 2.1. Synthesis of Co_3_O_4_ Bulk

Co_3_O_4_ bulk was synthesized by the thermal decomposition method. The detailed procedures were as follows: 1 g Co(NO_3_)_2_•6H_2_O was firstly calcined at 900 °C for 24 h in an air-filled furnace, and after cooling to room temperature, the final Co_3_O_4_ bulk was obtained.

### 2.2. Synthesis of Co_3_O_4_/Au Bulk Composites

The Co_3_O_4_/Au bulk catalysts were prepared by an impregnation method. The NaOH solution (0.1 M) was firstly added dropwise into the HAuCl_4_ solution (4 g/L) until pH = 6. Then, Co_3_O_4_ bulk was dispersed in the above solution. The composites were formed after being dried and calcined at 400 °C for 30 min.

### 2.3. Synthesis of Au Nanoparticles

Au nanoparticles were synthesized through the Frens’s approach [18]. In a typical procedure, a trisodium citrate solution (4 mL, 1 wt%) was firstly injected rapidly into a boiling HAuCl_4_ aqueous solution (100 mL, 0.01 wt%) under vigorous stirring. After refluxing for 15 min., the product was cooled to room temperature, and the purple transparent solution was obtained.

### 2.4. Synthesis of Co_3_O_4_ Ultrathin Porous Nanosheets

The ultrathin porous Co_3_O_4_ nanosheets were synthesized by a previously reported procedure [19]. Firstly, polyethylene oxide–polypropylene oxide–polyethylene oxide (PEO-PPO-PEO, Pluronic (P123, 0.2 g) was dissolved in the mixture of 0.8 g H_2_O and 13 g ethanol. Secondly, 0.125 g Co(Ac)_2_•4H_2_O and 0.07 g hexamethylenetetramine (HMTA) were added, and a purple solution formed after stirring for 15 min. Subsequently, 13 mL ethylene glycol (EG) was added, and a transparent solution was obtained after continuous stirring for 30 min. After aging for 1 day, the obtained solution was transferred into a 45 mL autoclave for solvothermal treatment at 170 °C for 2 h. The resultant product was washed for several times by ethanol and water, and Co_3_O_4_ nanosheets were collected after freeze drying and heated at 300 °C for 0.5 h.

### 2.5. Synthesis of Co_3_O_4_/Au Nanocomposites 

After being washed with deionized water, the as-obtained Au solution (25 mL) was mixed with ultrathin porous Co_3_O_4_ nanosheets (15 mg) in water. Then, the obtained suspension was stirred for 2 h. Finally, the brown powders were collected after freeze drying.

### 2.6. Characterization

The composition and crystallinity of the samples were characterized by X-ray diffraction (XRD, Rigaku-TTR III X-ray diffractometer, Tokyo, Japan) employing Cu-Kα radiation (λ = 1.5405 Å). Transmission electron microscopy (TEM) measurements were conducted on a JEOL-JEM 2010 equipment. Scanning transmission electron microscopy high-angle annular dark field (STEM-HAADF) was carried out on a JEM-ARM200F high-resolution transmission electron microscopy (HRTEM, JEOL, Tokyo, Japan). Scanning electron microscopy (SEM, JEOL, JSM-6700F, Japan) was employed to characterize the morphologies of the samples. The thickness of nanosheets was observed by atomic force microscope (AFM, MultiMode V, Veeco, Plainview, NY, USA). The X-ray photoelectron spectroscopy (XPS) of samples was collected on a Thermo ESCALAB 250 with Al-Kα radiation (ThermoFisher Scientific, Waltham, MA, USA). The soft X-ray absorption near edge-structure (XANES) spectra for Co L-edge and O K-edge was conducted at the BL12B-a Line of National Synchrotron Radiation Laboratory, China, and the total electron yield (TEY) mode was adopted. SPEX-1403 laser Raman spectrometer (SPEX, Costa Mesa, CA, USA) was employed to measure the Raman spectra of the samples.

All the electrochemical measurements were conducted in a three-electrode system on an electrochemical workstation (CHI instruments 660E, Shanghai City, China). Firstly, 5 mg catalysts were dispersed in a mixed solvent of water/isopropanol with a volume ration of 3:1 (1 mL). Subsequently, an 80 μL Nafion solution was added, and homogeneous ink formed after sonication for 1 h. Then, 3 μL dispersion was loaded on the glass carbon working electrode, and Pt foil and Ag/AgCl (3.5 M KCl) were adopted as the counter electrode and the reference, respectively. Linear sweep voltammetry (LSV) was carried out in 0.1 M KOH solution to obtain the polarization curves. The cycling stability was characterized by cyclic voltammetry (CV) from 0 to 1.0 V (vs. Ag/AgCl). The electrochemical impendence spectra (EIS) were measured at 0.5 V (vs. Ag/AgCl) with a frequency ranging from 1 Hz to 1000 kHz. The measured potential values vs. Ag/AgCl were converted to RHE according to the formula E (RHE) = 0.205 + E (vs. Ag/AgCl) + 0.059 pH.

## 3. Results and Discussion

Figure 1 exhibits XRD patterns of Co_3_O_4_ samples with or without Au. The diffraction peaks of the sample without Au loading are indexed to the standard spinel-structured Co_3_O_4_ (JCPDS File No. 65-3103) [20], indicating that cobalt precursors have been completely transformed to the Co_3_O_4_ phase. After decorating with Au NPs, the (111) peak of metallic Au (JCPDS No. 65-8601) is observed [21], suggesting the successful syntheses of Co_3_O_4_/Au nanocomposites by simply mixing Co_3_O_4_ with Au nanoparticles. Moreover, Co_3_O_4_ bulk and the corresponding Co_3_O_4_/Au bulk composites are also prepared successfully for comparison Appendix A.

The as-prepared samples are further examined by SEM, TEM, and AFM. The SEM image in Figure 2a shows that the wrinkles and transparent Co_3_O_4_ nanosheets are obtained, which have a size of 2–5 μm. Moreover, the porous structure with a size of ~5 nm is also formed in Co_3_O_4_ nanosheets after calcining at 300 °C for 0.5 h (see HRTEM images in S3-5), suggesting the successful synthesis of ultrathin porous Co_3_O_4_ nanosheets. HAADF-STEM is also conducted to further affirm the structure of Co_3_O_4_ nanosheets (Figure 2b), where the bright points represent Co atoms. The selected area electron diffraction (SAED) analysis shows that the nanosheets could be indexed to the (111) plane of cubic spinel Co_3_O_4_. The nonlinear arrangement of Co atoms in the HAADF image (Figure 2b) suggests the slight lattice distortion in Co_3_O_4_ nanosheets due to the ultrathin thickness and porous structure. As shown in Figure 2c,d, the height of nanosheets is measured by AFM, which is approximate 2.6 nm, about three times higher than 0.8 nm, the unit-cell dimension of cubic Co_3_O_4_ (JCPDS No. 65-3103). Moreover, the morphology and detailed microstructure of Co_3_O_4_/Au nanocomposites are also investigated by TEM and HRTEM. From Figure 2e, it can be clearly observed that Au nanoparticles (black points) are loaded successfully onto the surface of Co_3_O_4_ nanosheets, and the closest lattice fringes of Au (the red circle in Figure 2e) are ~0.23 nm, corresponding to the (111) lattice plane of Au. Further, STEM-EDS elemental mapping is used to characterize the elemental distribution of Co_3_O_4_/Au nanocomposites (Figure 3), and the elemental mapping images of the Co, O, and Au elements (Figure 3b,d) suggest that Au nanoparticles are loaded onto Co_3_O_4_ nanosheets uniformly.

Having studied the structure features of ultrathin porous Co_3_O_4_ nanosheets and Co_3_O_4_/Au nanocomposites, we investigate the OER activities of nanocatalysts in 0.1 M KOH (experimental section). Among Co_3_O_4_/Au nanocomposites with different Au loadings (10 mL, 15 mL, 20 mL, 25 mL, and 30 mL), the Co_3_O_4_/Au (15 mg/25 mL) sample shows the best OER performance (Appendix A). Figure 4a is the LSV curves of Au nanoparticles, IrO_2_, ultrathin Co_3_O_4_ nanosheets, and Co_3_O_4_/Au nanocomposites (15 mg/25 mL) in O_2_-saturated 0.1 M KOH with a scan rate of 5 mVs^−1^. Au nanoparticles show little OER performance, indicating that Au nanoparticles are OER inactive. After decorating Au nanoparticles onto Co_3_O_4_ ultrathin porous nanosheets, Co_3_O_4_/Au nanocomposites exhibit significant improvement of OER performance compared with Co_3_O_4_ nanosheets. Additionally, it is also worth noting that both Co_3_O_4_/Au nanocomposites and Co_3_O_4_ nanosheets display superior OER activities to commercial IrO_2_. Moreover, it is important to compare the overpotential required for getting a current density of 10 mA cm^−2^, which corresponds to the significant level of high efficiency of the solar water splitting system [22,23]. At the current density of 10 mA cm^−2^ (Figure 4b), Co_3_O_4_/Au nanocomposites display a smaller overpotential (0.46 V) than Co_3_O_4_ nanosheets (0.51 V) and commercial IrO_2_ (0.54 V). Further, the current density of ultrathin porous Co_3_O_4_ nanosheets and Co_3_O_4_/Au nanocomposites at an overpotential of 0.5 V (1.73 V vs. RHE) is also compared in Figure 4b. With the loading of Au nanoparticles, the current density is enhanced from 9.97 mA cm^−2^ for Co_3_O_4_ nanosheets to 14.76 mA cm^−2^ for Co_3_O_4_/Au nanocomposites, much higher than 0.73 mA cm^−2^ and 1.33 mA cm^−2^ of bulk Co_3_O_4_ and bulk Co_3_O_4_/Au (Appendix A), respectively. Furthermore, the corresponding Tafel plots in Figure 4c suggest that Co_3_O_4_/Au nanocomposites possess smaller Tafel slopes (88 mV dec^−1^) than Co_3_O_4_ ultrathin nanosheets (100 mV dec^−1^), meaning that Au loading onto Co_3_O_4_ ultrathin porous nanosheets accelerates the catalytic process of the OER. It can be concluded that Au nanoparticles could enhance the OER performance of ultrathin porous Co_3_O_4_ nanosheets, which is firstly observed in the composites obtained by physically mixed Co_3_O_4_ nanosheets with Au nanoparticles. The possible reason is that ultrathin nanosheets possess more low-coordinated surface atoms than bulk materials [13], meaning that the interaction between the physically mixed Co_3_O_4_ ultrathin nanosheets and Au nanoparticles could happen more easily. To further comprehend the kinetics process for the OER, the electrochemical impendence spectra (EIS) data are fitted using an equivalent circuit diagram consisting of the charger transfer resistance (R_ct_), the solution resistance (R_S_), and the constant phase element (CPE) related to the electrochemical double capacitance. The R_ct_ is related to the kinetics process of the OER. Figure 4d shows that the R_ct_ of Au-decorated nanosheets (73 Ω) is smaller than pure Co_3_O_4_ nanosheets (78 Ω), demonstrating the faster charge transfer kinetics during electrochemical reaction. Furthermore, the polarization curve slightly decreases after 1000 cycles, revealing excellent durability of Co_3_O_4_/Au nanocomposites, as shown in Figure 4e.

To reveal the role of Au nanoparticles in enhancing the OER performance of ultrathin porous Co_3_O_4_ nanosheets, the corresponding XPS and Raman spectra were conducted. As shown in Figure 5a, compared with ultrathin porous Co_3_O_4_ nanosheets, Co_3_O_4_/Au nanocomposites show a ~0.7 eV decreased electron binding energy for Co 2P_3/2_ and ~0.4 eV decreased electron binding energy for 2P_1/2_. The binding energy shift is due to the increased electron density around Co atoms in Co_3_O_4_/Au nanocomposites [17], which indicates the interaction between the nanosheets and Au loading. To further verify the above speculation that the interaction happens between the physically mixed Co_3_O_4_ ultrathin nanosheets and Au nanoparticles, the Raman measurement is conducted (Figure 5b). Five peaks at 671, 606, 510, 471, and ~188cm^−1^ are observed in both the Raman spectra of the Co_3_O_4_/Au nanocomposites and Co_3_O_4_ nanosheets, which correspond to A_1g_, F_2g_, F_2g_, E_g,_ and F_2g_ modes of the Co_3_O_4_ spinel structure, respectively [24]. Compared with the nanosheets, the Raman peaks of the Co_3_O_4_/Au nanocomposites exhibit a blue shift, and the peak at 671 even shifts to 687cm^−1^ by an increase of 18 cm^−1^, which further certifies the electronic interaction between ultrathin Co_3_O_4_ porous nanosheets and Au nanoparticles [17].

To get an in-depth understanding of the electronic interaction between the Co_3_O_4_ ultrathin nanosheets and Au nanoparticles and the corresponding effect on the OER performance of the Co_3_O_4_/Au nanocomposites, XANES is also performed, which is sensitive to the electronic states of the special elements. Figure 5c shows the normalized Co L-edge XANES for the Co_3_O_4_ nanosheets and the Co_3_O_4_/Au nanocomposites, along with the reference samples CoO and LaCoO_3_ [25]. The Co L-edge XANES spectra correspond to the excitation of 2p electron to the unoccupied 3d state. Additionally, for the Co L_3_-edge spectrum in Co_3_O_4_, there are two peaks at 779.5 eV and 777.6 eV, originating from the octahedrally coordinated Co^3+^ and tetrahedrally coordinated Co^2+^, respectively. With the loading of Au nanoparticles onto ultrathin Co_3_O_4_ nanosheets, the peak at 777.6 eV becomes stronger, while the peak at 779.5 eV weakens, indicating that the ratio of Co^2+^/Co^3+^ increases in Co_3_O_4_ nanocomposites with the loading of Au nanoparticles. Furthermore, the O K-edge XANES spectra of Co_3_O_4_/Au nanocomposites (Figure 5d) show an obvious decrease in the unoccupied O 2p-Co 3d hybridized state (~529.8 eV) compared with that of the Co_3_O_4_ nanosheets, suggesting a higher electronic density at the O sites, which strengthens the covalent Co-O bonding in the Co_3_O_4_/Au composite, thus improving the conductivity of the nanocomposites [26], which is consistent with the EIS results. It has been well-documented that Co^2+^ can release electrons under an applied bias, thus promoting an affinity to oxygen ions on the surface of the catalyst to form CoOOH, i.e., the main active site for OER [27,28,29,30,31]. In conclusion, due to the addition of Au nanoparticles, the increased Co^2+^/Co^3+^ ratio and conductivity result in the enhanced OER performance of the Co_3_O_4_/Au nanocomposites. For comparison, the bulk Co_3_O_4_ and bulk Co_3_O_4_/Au composites are also prepared, and no obvious OER enhancement is observed with the loading of the Au nanoparticles (Appendix A). Compared with the formed reports [32,33,34,35], the ultrathin porous Co_3_O_4_ nanosheets do not exhibit superior electrocatalytic properties, which could be due to the low electron conductivity and imperfect synthesis process of the Co_3_O_4_ nanosheets.

## 4. Conclusions

In summary, ultrathin porous Co_3_O_4_ nanosheets are synthesized successfully by a facile method. The thickness of the nanosheet is about three times that of the unit-cell dimension (0.8 nm) given for Co_3_O_4_. Additionally, the corresponding OER performance is firstly enhanced by physically mixing Co_3_O_4_ ultrathin porous nanosheets with Au nanoparticles, which originated from the electronic interaction between Co_3_O_4_ and Au nanoparticles due to the unsaturated bonds on the surface of Co_3_O_4_ ultrathin porous nanosheets. With the loading of Au nanoparticles, the current density of ultrathin porous Co_3_O_4_ nanosheets is enhanced from 9.97 to 14.76 mA cm^−2^ at an overpotential of 0.5 V. Meanwhile, Co_3_O_4_/Au nanocomposites also exhibit a lower overpotential and a Tafal slope, as well as good stability. The enhanced OER performance is ascribed to a higher Co^2+^/Co^3+^ ratio and quicker charge transfer in the Co_3_O_4_/Au nanocomposites than in the ultrathin porous Co_3_O_4_ nanosheets, and Co^2+^ ions are beneficial to the formation of CoOOH, which acts as the main active site for OER.

## Figures and Tables

**Figure 1 nanomaterials-12-04419-f001:**
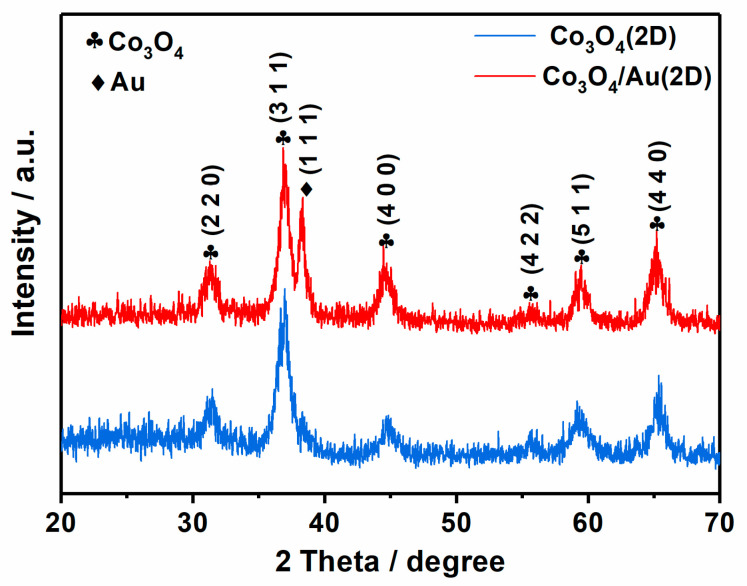
XRD patterns of Co_3_O_4_ nanosheets and Co_3_O_4_/Au nanocomposites.

**Figure 2 nanomaterials-12-04419-f002:**
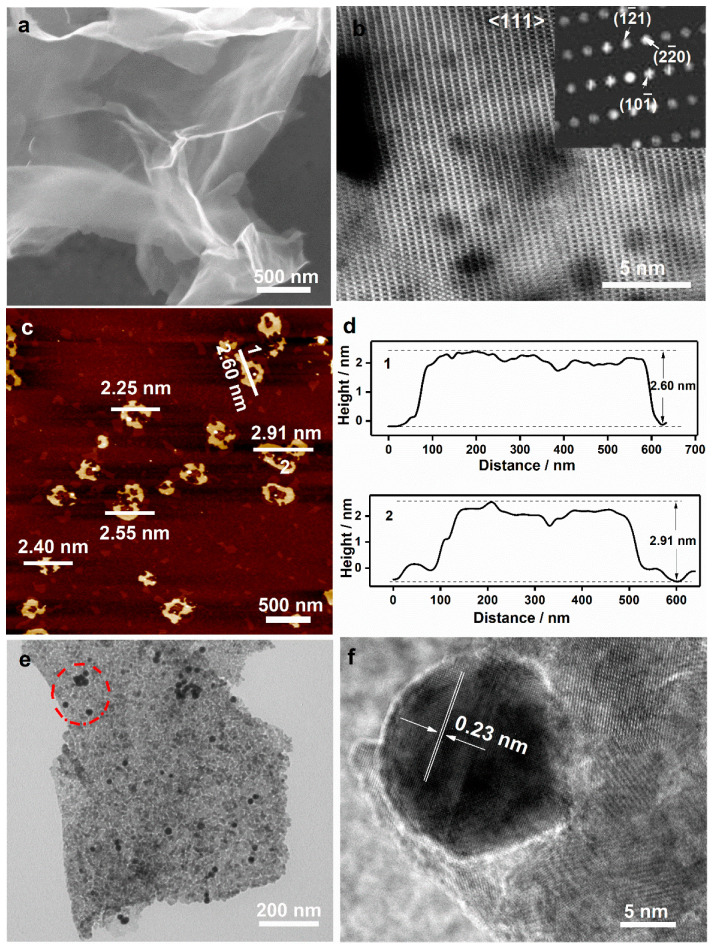
(**a**) SEM and (**b**) STEM-HAADF images of Co_3_O_4_ ultrathin nanosheets; (**c**) AFM image and (**d**) the corresponding height profiles of Co_3_O_4_ ultrathin nanosheets; and (**e**) TEM and (**f**) HRTEM images of Co_3_O_4_/Au nanocomposites.

**Figure 3 nanomaterials-12-04419-f003:**
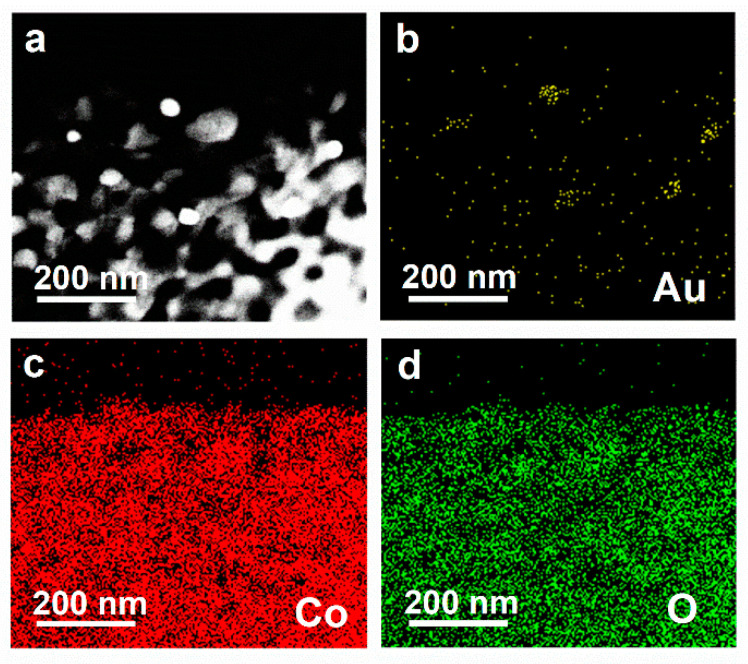
(**a**) Dark-field STEM micrograph of Co_3_O_4_/Au nanocomposites; and STEM-EDS elemental mapping of (**b**) Au, (**c**) Co, and (**d**) O elements.

**Figure 4 nanomaterials-12-04419-f004:**
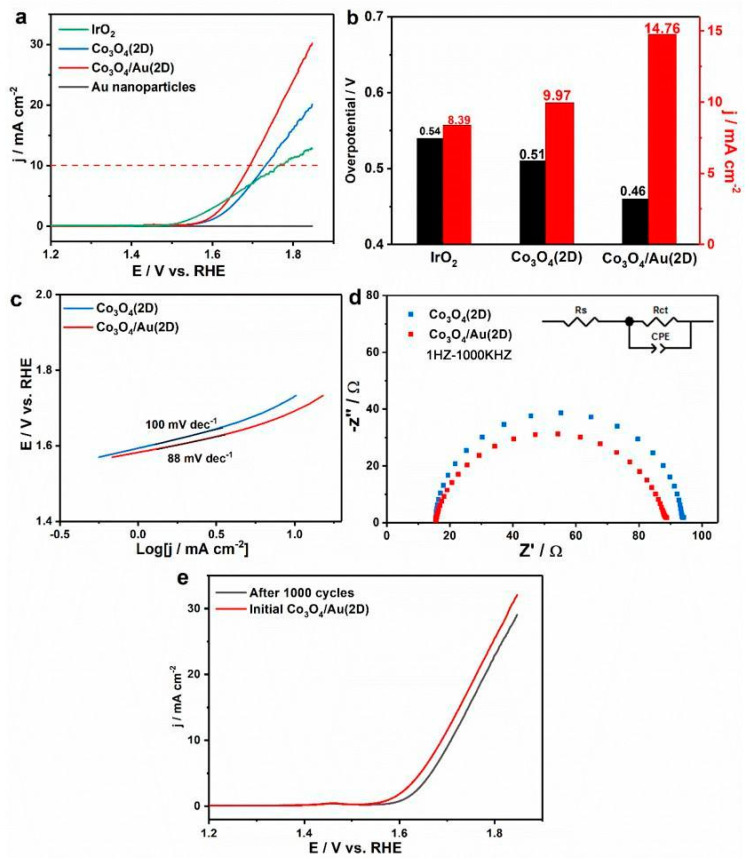
OER activities of the nanocatalysts: (**a**) LSV curves of Au nanoparticles, IrO_2_, Co_3_O_4_ ultrathin nanosheets, and Co_3_O_4_/Au nanocomposites; (**b**) comparison of the current density at an overpotential of 0.5 V and the overpotential required for achieving a current density of 10 mA cm^−2^, for IrO_2_, Co_3_O_4_ ultrathin porous nanosheets, and Co_3_O_4_/Au nanocomposites; (**c**) Tafel plots for Co_3_O_4_ nanosheets and Co_3_O_4_/Au nanocomposites; (**d**) EIS of Co_3_O_4_ nanosheets and Co_3_O_4_/Au nanocomposites; and (**e**) LSV curves of the Co_3_O_4_/Au nanocomposites before and after 1000 cycles.

**Figure 5 nanomaterials-12-04419-f005:**
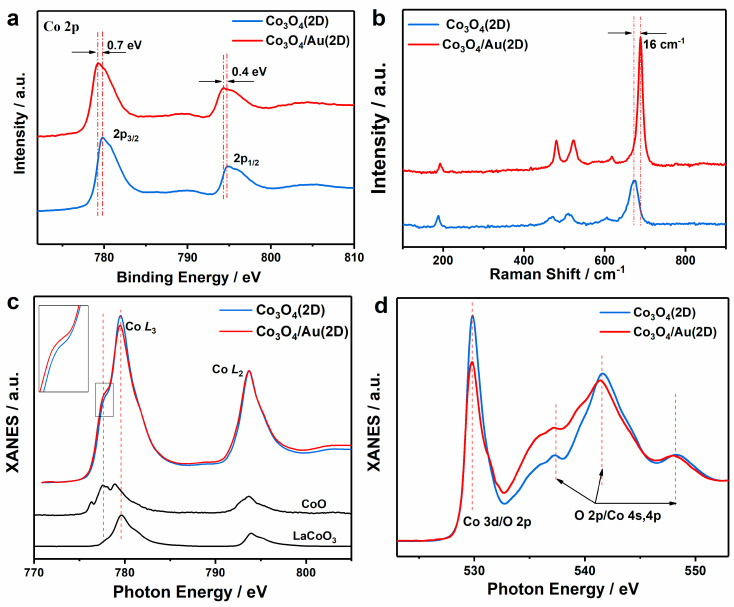
The electronical structure characterizations of Co_3_O_4_ ultrathin nanosheets and Co_3_O_4_/Au nanocomposites: (**a**) XPS spectra of Co 2p level; (**b**) Raman spectra between 100 and 900 cm^−1^; (**c**) XANES spectra of Co L-edge in Co_3_O_4_ ultrathin nanosheets and Co_3_O_4_/Au nanocomposites, and XANES spectra of LaCoO_3_ and CoO are cited from Ref [25]; and (**d**) XANES spectra of O K-edge in Co_3_O_4_ ultrathin nanosheets and Co_3_O_4_/Au nanocomposites.

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
