# Peer review of "Enhanced Electrocatalytic Water Oxidation of Ultrathin Porous Co3O4 Nanosheets by Physically Mixing with Au Nanoparticles"

_nanomaterials, 2022, doi:10.3390/nano12244419_

Round 1

Reviewer 1 Report

For the last two decades the catalytic water oxidation (oxygen evolution reaction) remains a hot area in chemistry. Thousands of papers have been published on different aspects of this reaction. This area and experimental methods including quantification of dioxygen yield are very well developed. However, I have noticed the significant increase of number of publications with the titles containing wording “water oxidation” or “oxygen evolution” without any proof of oxygen formation. Now, the measurements of dioxygen have begun an obligatory in this field. This is exceptionally important for studies of electrocatalytic systems, where the increase of the current is interpreted as water oxidation. The formation of dioxygen was neither proved in this work nor in the similar systems (refs 13-15). For this reason, I do not recommend this manuscript for publication unless the authors perform additional experiments and prove that dioxygen is really formed in the reported system.  I would not accept the argument that  if some papers are published without evidence of dioxygen formation, this manuscript can be also published.

Reviewer 2 Report

Dear authors

your article presents interesting results and well commented, I have three small remarks before validating the publication of your work

- Can you describe part 2.1 or put a reference?

- figure 4.d: can you insert the imposed frequencies on your impedance diagram, this will allow the reader to see the imposed frequency and to define reactions,

- for the bibliography, can you compare your results with literature other than Chinese references from groups in Europe or the USA working in this field and you do not cite them

thank you in advance for taking my comments into account.

reviewer 1

Reviewer 3 Report

In this manuscript, the significant enhancement of Co3O4 nanosheets activity towards the oxygen evolution reaction thanks to incorporation of Au nanoparticle was revealed, what could be used for efficient water electrolysis. The results are reliable and the investigation is interesting.

Minor spell check required, e.g.:

line 100

The as-obtained Au solution (25 mL) was washed, and then mixed with Co3O4 ultrathin porous nanosheet power (15 mg) in water.

Round 2

Reviewer 1 Report

I would be happy to recommend this manuscript for publication as soon as the authors "perform additional experiments to prove the formation of dioxygen in the system." So far there is no evidence for dioxygen formation.